# Analysis of Dietary Patterns Associated with Kidney Stone Disease Based on Data-Driven Approaches: A Case-Control Study in Shanghai

**DOI:** 10.3390/nu16020214

**Published:** 2024-01-09

**Authors:** Yifei Wang, Shaojie Liu, Qi Zhao, Na Wang, Xing Liu, Tiejun Zhang, Gengsheng He, Genming Zhao, Yonggen Jiang, Bo Chen

**Affiliations:** 1Department of Nutrition and Food Hygiene, School of Public Health, Fudan University, Shanghai 200032, China; wangyf22@m.fudan.edu.cn (Y.W.); gshe@shmu.edu.cn (G.H.); 2Department of Clinical Nutrition, The First Affiliated Hospital of Xiamen University, School of Medicine, Xiamen University, Xiamen 361003, China; liushaojie@fudan.edu.cn; 3Key Lab of Public Health Safety of the Ministry of Education, School of Public Health, Fudan University, Shanghai 200032, China; zhaoqi@shmu.edu.cn (Q.Z.); na.wang@fudan.edu.cn (N.W.); liuxing@fudan.edu.cn (X.L.); tjzhang@shmu.edu.cn (T.Z.); gmzhao@shmu.edu.cn (G.Z.); 4Department of Epidemiology, School of Public Health, Fudan University, Shanghai 200032, China; 5Songjiang District Center for Disease Control and Prevention, Shanghai 201620, China; sjjktj@hotmail.com

**Keywords:** dietary pattern, kidney stone, PCA, LASSO regression

## Abstract

The main objective of this study was to analyze dietary patterns using data-driven approaches and to explore preventive or risk dietary factors for kidney stone disease (KSD). A case–control matching study was conducted in adults (*n* = 6396) from a suburb of Shanghai. A food frequency questionnaire was used to assess the consumption of various types of food, and B-ultrasound was used to identify kidney stones. Principal component analysis and regression were used to generate dietary patterns and further explore the relationship between dietary patterns and KSD. LASSO regression and post-selection inference were used to identify food groups most associated with KSD. Among males, the “balanced but no-sugary-beverages pattern” (OR = 0.78, *p* < 0.05) and the “nuts and pickles pattern” (OR = 0.84, *p* < 0.05) were protective dietary patterns. Among females, “high vegetables and low-sugary-beverages pattern” (OR = 0.83, *p* < 0.05) and “high-crustaceans and low-vegetables pattern” (OR = 0.79, *p* < 0.05) were protective dietary patterns, while the “comprehensive pattern with a preference for meat” (OR = 1.06, *p* < 0.05) and “sugary beverages pattern” (OR = 1.16, *p* < 0.05) were risk dietary patterns. We further inferred that sugary beverages (*p* < 0.05) were risk factors and pickles (*p* < 0.05) and crustaceans (*p* < 0.05) were protective factors.

## 1. Introduction

With the complexity of dietary structures and continuous improvements in life conditions, the incidence and postoperative recurrence rate of kidney stone disease (KSD) have been increasing annually. KSD is currently a global health care issue affecting people in almost all developed and developing countries [1,2]. Despite the availability of effective treatments for KSD, the high rate of recurrence after stone removal still has a profound impact on the individual workforce and family life of patients. Furthermore, it imposes severe health and economic burdens on society. China has become one of the three regions in the world with the highest incidence of KSD [3]. A large-scale cross-sectional study conducted in 2017 showed that the prevalence of adult KSD in China is approximately 6.5% for males and 5.1% for females, affecting more than 60 million patients [4].

The formation of stones is strongly related to factors such as age, sex, diet, environment, genetics, urinary anatomy and histological abnormalities, infection, etc. [5]. Therefore, KSD has come to be recognized as a chronic metabolic disease requiring multidisciplinary involvement. Dietary factors play an important role in the formation of kidney stones. Nutrient intake is closely related to the components of urine excretion, so poor eating habits can lead to metabolic disorders such as hypercalciuria, hyperoxaluria, hypocitraturia, and hyperuricosuria [6,7,8].

In terms of certain nutrient intake, insufficient intake of fluids and calcium and excessive intake of animal protein, vitamin C, sugar and purine have been reported to be risk factors for KSD [9,10,11,12,13,14]. While considering a certain food or food group, researchers found that meat is a risk factor, while citrus fruits, caffeinated beverages (tea and coffee), and whole grains are protective factors [14,15,16]. In recent decades, epidemiological studies of diet and disease have focused on linking dietary patterns to disease or disease risk rather than individual nutrients or foods, as key nutrients co-exist and interact [17]. Some dietary patterns have been found to be associated with prevention or risk of KSD. The Mediterranean dietary pattern, defined as high consumption of plant-based foods containing monounsaturated to saturated fatty acids and low consumption of meat, is associated with a lower risk of KSD [18,19]. And lacto-ovo-vegetarianism (characterized by high content of fruits and vegetables and a balanced intake of low-fat dairy products) may be beneficial for KSD prevention [20]. Other studies found that diet with a high dietary inflammatory index (DII) is significantly related to the increased risk of having hypercreatinuria, hypercalciuria, hyperuricosuria, and hypocitraturia in stone former men [21,22,23,24]. These dietary patterns were obtained through hypothesis-driven approaches. But existing prior knowledge may not fully explain the cause of KSD. And as people’s diets become more complex, prior knowledge may not adequately represent the real diet of the study subjects. Therefore, it is necessary to apply data-driven methods to analyze dietary patterns of KSD patients; however, few researchers have used such methods to generalize dietary patterns and explore their association with KSD, especially in the Chinese population. Recently, researchers have applied advanced statistical data-mining techniques to the concept of dietary patterns. These techniques can better capture the inherent complexity of dietary intake data sets while incorporating many variables required for dietary pattern analysis [15,25,26,27], such as random forest (RF) and the LASSO (least absolute shrinkage and selection operator) model [28,29,30].

In the field of diet and KSD, there is a lack of practice in data-driven dietary pattern analysis methods; especially in recent years, few high-quality large-scale population studies have been conducted in the Chinese population. We designed a case–control study in a large community population from Shanghai, China, using traditional and emerging data-driven approaches to analyze dietary patterns and identify key factors associated with the risk or prevention of KSD.

## 2. Method

### 2.1. Data Set

The Shanghai Suburban Adult Cohort and Biobank (SSACB) was established to identify environmental, lifestyle and genetic risk factors for non-communicable chronic diseases (NCDs) in adults (20–74 years old) living in a suburban area of Shanghai with rapid urbanization. The subjects in this study were respondents who participated in the SSACB baseline survey in 2017. The details of SSACB have been described previously [31]. In brief, the baseline data on demographic, lifestyle and physical health-related factors were collected using a face-to-face questionnaire interview, and all participants completed physical examinations and urine routine. Physical examination was performed by professional doctors from district community health service centers. Urine routines were performed in the laboratory of the hospital.

Cases of kidney stones are defined by B-ultrasound. If the B-ultrasound results reported stones or crystals, the participant was selected as a case in this study. During the face-to-face interview, all participants completed a validated, semi-quantitative Food Frequency Questionnaire (FFQ), adapted from an existing FFQ used in the Shanghai Diet and Health Study since 2010, which contained 29 food groups composed of similar food items, representing approximately 95% of the most commonly consumed foods in Shanghai [32]. The reliability and validity of the questionnaire have been verified [33]. Urine pH, water intake, and tea consumption, which are considered to be associated with the formation of kidney stones, were included as covariates. Other covariates include general population information and lifestyle such as gender, age, education, smoking, alcohol use, etc. Smoking was defined as having smoked at least one cigarette per day for six months or longer. Alcohol use was defined as drinking at least three times per week for six months or longer. Tea consumption was also defined as drinking tea at least three times per week for six months or longer.

### 2.2. Statistical Analysis

In our preliminary analysis, we found that the above covariates were very unevenly distributed between cases and non-cases (Appendix A) and that the prevalence of KSD was low in the study population. We adopted a case–control matching (CCM) design to control confounding, improve statistical power, and avoid the influence of extreme cases. We used the nearest neighbor principle to match 5 controls for each case, and the covariates included in the calculation are age, BMI, education, smoking, alcohol use, water intake, tea consumption, and urine pH. All statistical analysis were performed separately for males and females.

### 2.3. Principal Component Analysis (PCA) and Logistic Regression with Selected Principle Components (PCs)

PCA was used to identify major dietary patterns among participants based on their food intake data. PCs were rotated using varimax rotation to improve their interpretability. Factor loadings greater than 0.3 were considered as high and less than −0.3 as low, respectively. Dietary patterns were then named according to high and low factor loadings. The orientation of the loadings represents the propensity to eat or not to eat a certain group of foods, and the absolute value of the loadings represents the amount of food consumed. Top PCs were extracted based on the scree plot, eigenvalues and cumulative proportion of total variance. These selected top PCs were then included in a multivariate logistic regression analysis to explore the relationship between dietary patterns and KSD.

### 2.4. Least Absolute Shrinkage and Selection Operator (LASSO) Regression and Post-Selection Inference

LASSO regression was used to identify dietary patterns associated with KSD among participants based on their food intake data. The optimal value of lambda was selected using 10-fold cross-validation, and the model with the minimum mean squared error was selected. The orientation and magnitude of the coefficients represent the direction and strength of the effect of each variable on the outcome. Further, important dietary factors with non-zero coefficients were included in a multivariate logistic regression model to perform post-selection inference.

In all analysis, the aforementioned covariates were adjusted in the model. All statistical analyses were performed using R version 4.2.3. Two-sided *p* values <0.05 were considered statistically significant.

## 3. Results

### 3.1. Personal Characteristics and Food Intake

Of the total sample of 46,658 participants, 1066 were diagnosed as KSD cases, of which 465 were male and 601 were female, and the prevalence of KSD was 2.46% and 2.16% for males and females, respectively. Some covariates are very unevenly distributed between cases and non-cases (see Appendix A for details); for example, the mean age of cases is more than 9 years younger than that of non-cases in both males and females, the education level of cases is generally higher than that of non-cases, and the BMI of cases is lower than that of non-cases in females. There was no significant difference in water intake among males, females, or the total population, and the distribution was similar across the common range of water intake (1000–1500 mL, 1500–2000 mL, 2000–3000 mL) and the more extreme range of water intake (below 1000 mL or above 3000 mL). There was no difference in urine pH between cases and non-cases in males, while urine pH was higher in females with stones. After 1:5 case–control matching, the covariates were uniformly distributed among the case and control groups.

As shown in Table 1, among males, the case group consumed more fruits, dairy, soymilk, carbonated drinks, pure juices, other sugary drinks, fried dough foods, and less pickles. Among females, the case group consumed more wheat, mushrooms, pork, meat from other livestock, poultry, dairy, yogurt, soymilk, carbonated drinks, pure juices, other sugary drinks, candies and chocolates, and fried dough foods.

### 3.2. Principal Component Analysis (PCA) and Logistic Regression with Selected Principal Components

Among males, PCA yielded nine PCs that together accounted for 50% of the total variation, with PC1 accounted for 16.4%. To interpret the dietary patterns represented by these PCs, we focused on PC loadings with absolute values greater than 0.3. For example, PC1 (dietary pattern 1) has high positive loadings of meats (e.g., poultry and livestock meats) and crustaceans, indicating that participants who followed the dietary pattern represented by PC1 typically consumed greater amounts of these foods. So, we named PC1 the “meat pattern”. PC2 is defined as a dietary pattern that represents high intake of aquatic products (e.g., crustaceans, and freshwater fish) and low intake of sugary beverages (carbonated drinks, other sugary drinks and pure juices). PC3 characterizes a dietary pattern with low consumption of fresh vegetables and dark vegetables. PC4 represents a dietary pattern that has high loadings of processed meat and sugary drinks (carbonated and other sugary drinks), and low loadings of dairy, yogurt and fruits. The remaining PCs can be interpreted in a similar way based on the sign and absolute value of the PC loadings (Table 2 and Appendix A). The “low-sugary-beverages pattern” (PC2, OR = 0.78, *p* < 0.001) and “pickles and nuts pattern” (PC5, OR = 0.84, *p* < 0.001) are protective factors for KSD, while the “meat pattern” (PC5, OR = 1.14, *p* < 0.001) is a risk factor.

In contrast, females have more balanced dietary patterns. There are many food groups with absolute loading values between 0.2 and 0.3 distributed across PCs (Appendix A). As Table 2 shows, PC1 (OR = 1.07, *p* = 0.001) is a typical balanced dietary pattern, and regression analysis suggests that it is a risk factor for KSD. Another risk factor is PC4 (OR = 1.16, *p* = 0.003), which is characterized by high consumption of sugary beverages and low consumption of whole grains and mixed beans, and PC8 (OR = 1.12, *p* = 0.014), which is characterized by high consumption of rice and rice products. The “high-vegetables and low-sugary-beverages pattern” (PC2, OR = 0.88, *p* < 0.001) and the “high-crustaceans and low-vegetables pattern” (PC5, OR = 0.79, *p* < 0.001) are protective factors.

### 3.3. Least Absolute Shrinkage and Selection Operator (LASSO) Regression and Post-Selection Inference

The LASSO model defined 5 and 15 food groups with non-zero coefficients helpful for predicting KSD in males and females, respectively (Table 3). For both males and females, pure juices, fried dough foods, carbonated drinks, other sugary drinks, and dairy are risk factors, pickles and crustaceans are protective factors. Meanwhile, for females, candies and chocolates, poultry, pork, meat from other livestock, soymilk, yogurt, and wheat are also risk factors. Multivariate logistic regression with selected important factors showed that for both males and females, other sugary drinks (for males, β = 2.3 × 10^−3^, *p* = 0.053; for females, β = 5.4 × 10^−3^, *p* = 0.053) are risk factors, while pickles (for males, β = −1.9 × 10^−2^, *p* < 0.001; for females, β = −7.6 × 10^−2^, *p* = 0.046) and crustaceans (for males, β = −9.2 × 10^−3^, *p* = 0.022; for females, β = −1.1 × 10^−2^, *p* = 0.002) are protective factors. Moreover, pure juices (β = 7.7 × 10^−3^, *p* < 0.001) and carbonated drinks (β = 1.6 × 10^−3^, *p* = 0.029) are risk factors for males and females, respectively.

## 4. Discussion

In this study, we used data-driven approaches to analyze dietary patterns and explore their association with KSD. PCA followed by further principal component regression revealed several dietary patterns associated with risk or prevention of KSD. Among males, the “low-sugary-beverages pattern” and the “nuts and pickles pattern” were protective dietary patterns. Among females, “high-vegetables and low-sugary-beverages pattern” and “high-crustaceans and low-vegetables pattern” were protective dietary patterns, while “comprehensive pattern with a preference for meat” and “sugary beverages pattern” were risk dietary patterns. We also used LASSO regression to select food groups most closely related to KSD. Extrapolating from post-selection inference, we concluded that among these food groups, sugary beverages were the most likely risk factor, while pickles and crustaceans were the most likely prevention factor. In today’s increasingly complex dietary environment, our findings provide new practical ideas for the prevention and treatment of KSD, especially in the high-risk population in contemporary Chinese dietary culture.

We found that sugary beverages (including pure juices, carbonated drinks, and other sugary drinks) are risk factors for KSD, and ours is the only study in recent years to find an association between sugary beverages intake and KSD in a Chinese population. Adequate fluid intake is the most important nutritional measure for preventing kidney stone recurrence [34], increasing urine dilution and thereby reducing the concentration of lithogenic components, and promotes crystal excretion by reducing tubular transport time [35,36]. But the effects of different beverages are still being debated. There is no consensus among researchers on the association between beverage intake and KSD [37,38,39]. The composition of beverages can affect the composition of urine, which can affect the tendency of stone formation. Both natural juices and artificially sweetened drinks are often rich in fructose, which may increase the excretion of calcium, oxalate and uric acid, and are associated with a higher risk of stone formation [40,41,42,43]. Ascorbic acid, existing naturally in fruits and vegetables or artificial added to beverages, can be converted to oxalate [35,44,45]. In addition, additives in beverages, such as artificial colors, preservatives, sweeteners, and mineral substances, severely limit the health benefits of these beverages for kidney stone formers [12,46]. However, Ferraro et al. concluded that orange juice reduced the risk of stone formation by 12% (*p* = 0.004) [39]. This may be due to the high content of citrate, which can affect crystallization by (a) increasing urine pH and then dissolving uric acid stones (however, the alkaline environment is conducive to the formation of calcium-containing stones); (b) inhibiting all processes of crystallization, including the inhibition of crystallization nucleation, growth, and aggregation; (c) forming a complex with urinary calcium to make a calcium-containing complex with high solubility, which reduces urinary calcium concentration and urinary calcium oxalate saturation [44,45,47]. When discussing the effects of fruit juice on stone formation in a population containing various types of stones, these promoting or inhibiting effects may cancel each other out. In short, whether the beverage promotes or inhibits the formation of stones is closely related to its specific ingredients. This suggests that patients with kidney stones should strictly limit the type of fluid while requiring more fluid intake in clinical practice.

We also found that pickles (a Chinese-style salted fermented vegetable) were protective factors for KSD, which is quite a novel finding. There is a limited association between pickles and KSD, but studies have shown that pickles contain probiotics, such as lactic acid bacteria that support gut health, boost the immune system, improve nutrient bioavailability, and reduce the risk of lactose intolerance, allergies, and certain malignancies [48,49]. In ecological and conventionally produced food processing, lactic acid fermentation (LAF) is widely used [50]. During fermentation, the concentration of many bioactive compounds increases, as does the bioavailability of iron, vitamin C, beta-carotene or betaine [51]. These factors may indirectly improve the metabolism of substances associated with lithiasis.

We also found that crustaceans (including shrimp, crabs and shellfish) are protective factors for KSD, which is somewhat different from previous experience. From the nutritional composition of the food, crustaceans are rich in protein, cholesterol and purines, and excessive intake of these nutrients can increase the risk of KSD; in particular, purines will increase uric acid in the urine and promote the formation of uric acid stones [52,53,54]. However, we found that eating crustaceans seems to be beneficial. We speculate that this may be related to the dietary habits of the Shanghai region from which the study subjects came. When people eat crustaceans, they often dip them in vinegar. Studies have found that vinegar prevents the formation of calcium oxalate (CaOx) stones in the kidney [55], possibly through (a) epigenetic regulation affecting urinary citrate and calcium excretion; (b) regulating intestinal flora and increasing blood acetate; (c) reduce CaOx crystal adhesion by restoring renal tubule cells [56,57,58]. This suggests that a thorough understanding of all the details of eating habits is necessary. In addition, since uric acid stones account for less than 10% of KSD patients in the Chinese population [3], the lithogenic effect of purine in seafood may not contribute much in the population.

The PCA method combined with further multiple logistic regression identified some dietary patterns associated with KSD, such as the dietary pattern of “more vegetables and less sugary beverages” in females as a protective factor. However, it is impossible to know whether eating more vegetables or drinking fewer sugary drinks plays the key protective role. So we introduced LASSO regression to help explain this complexity. Both LASSO and PCA are effective dimensionality reduction methods, but LASSO has another advantage that PCA does not have, that is, selection features. Feature selection refers to the selection of a small number of relevant features from the original features to help researchers better classify and regression the task, and try to eliminate irrelevant or redundant features [59]. In our study, this meant that the LASSO model more clearly identified the food groups most closely associated with KSD. Therefore, combined with the results of LASSO regression selection, we can speculate that the key protective factor of females’ “more vegetables and fewer sugary beverages” pattern may be low sugary beverage intake, rather than high vegetable intake.

Taken together, we identified some dietary patterns associated with KSD using principal component analysis and regression. We also applied the LASSO model and identified the food groups most strongly associated with KSD. We hypothesize that sugary beverages’ effect is primarily due to the presence of high sugars and additives that alter the metabolism of the lithogenic components and their eventual excretion through urine. The consumption of pickles and crustaceans may affect the metabolism of lithogenic components and renal repair after injury by affecting intestinal microbes. We also obtained similar results to previous studies, such as that meat (animal protein) intake is a risk factor for KSD. Of course, more studies are needed to confirm both our findings and our predictions.

### Strengths and Limitations

There are some strengths to this study. Firstly, although the relationship between diet and KSD is a well-studied topic, fewer studies have been conducted in Chinese populations in recent years. Our study found that sugary beverages (pure juices and sugary drinks) are risk factors for KSD, while pickles and crustaceans are protective factors, which have rarely been found in the Chinese population. Secondly, we newly used LASSO regression to analyze dietary data; this model is rarely applied in the field of nutritional epidemiology. However, there are some limitations in this study. The classification of food groups in the FFQ is not detailed enough to identify the food items that are specifically associated with KSD. For example, there is no further differentiation of juice types in “pure juices” group, which is not conducive to an in-depth investigation of juices that are more relevant to pathogenesis. Information on covariates, such as drug use that may affect metabolism, is not sufficiently comprehensive. In addition, we did not obtain a classification of stone components in patients, which is not conducive to exploring the mechanism by which food affects stone formation, because different types of stones may be affected differently by nutrients.

## 5. Conclusions

This study explored dietary patterns associated with KSD. Principal component regression yielded several risk or protective dietary patterns for males and females, respectively. LASSO regression has also been successfully applied to identify sugary beverages as risk factors for KSD, as well as protective factors such as pickles and crustaceans, which is the largest innovation of this study. There are some limitations to the FFQ classification of food groups used in this study. In addition, some covariates associated with KSD, such as the amount of tea consumed or the amount of caffeine consumed, and urinary stone-promoting factor/inhibitor concentration, were not quantified. Our study provides a new perspective on the dietary prevention and treatment of KSD, but more epidemiological studies and mechanism studies are needed to further explore, in particular, patients with different types of kidney stones, who need to be discussed separately.

## Figures and Tables

**Table 1 nutrients-16-00214-t001:** Comparison of food intake between cases and control (g/day).

Food Groups	Male			Female			Total		
	Case (*n* = 465)	Control (*n* = 2325)	*p*	Case (*n* = 601)	Control (*n* = 3005)	*p*	Case (*n* = 1066)	Control (*n* = 5330)	*p*
Rice	393.9	391.6	0.833	267.4	254.1	0.102	323.1	310.5	0.074
Wheat	35.4	32.2	0.123	40.0	32.1	0.013	38.0	33.5	0.025
Whole grains and mixed beans	15.1	15.2	0.964	17.4	16.4	0.439	16.4	16.2	0.852
Potatoes	20.7	21.2	0.729	21.3	20.6	0.593	21.1	20.8	0.771
Fresh vegetables	182.1	186.5	0.601	195.8	190.7	0.570	190.2	187.9	0.712
Dark vegetables	91.6	89.1	0.658	95.5	86.9	0.126	93.8	90.2	0.368
Mushrooms	18.8	18.3	0.689	22.5	18.7	0.017	20.9	18.3	0.013
Fruits	112.6	98.5	0.017	105.0	99.0	0.230	108.4	101.4	0.065
Dairy	58.3	48.2	0.032	61.4	46.8	0.001	60.0	48.0	<0.001
Yogurt	40.7	32.3	0.116	40.0	31.2	0.003	40.3	31.2	0.002
Pork	35.6	37.3	0.460	43.3	35.4	0.004	40.1	36.8	0.078
Meat from other livestock meat	12.0	9.2	0.098	12.9	9.6	0.005	12.5	9.8	0.006
Poultry	18.7	15.9	0.123	19.3	15.8	0.004	19.0	15.8	0.002
Innards	4.2	3.6	0.190	5.2	3.6	0.054	4.8	3.5	0.010
Freshwater fish	26.2	26.3	0.946	26.6	27.3	0.629	26.5	26.8	0.721
Marine fish	12.6	13.3	0.473	12.5	12.3	0.841	12.6	12.5	0.918
Crustacean	17.3	16.6	0.754	14.8	16.4	0.079	15.9	16.5	0.621
Soymilk	23.9	16.6	0.010	29.1	16.7	<0.001	26.8	18.8	<0.001
Tofu	17.9	17.9	0.982	18.5	17.7	0.465	18.3	18.1	0.822
Eggs	29.9	29.6	0.848	28.8	29.4	0.596	29.3	29.2	0.892
Nuts	9.7	11.1	0.085	11.2	12.5	0.177	10.5	11.2	0.278
Carbonated drinks	31.8	15.9	0.001	30.6	13.9	<0.001	31.5	16.5	<0.001
Pure juices	16.8	7.2	<0.001	13.8	7.2	<0.001	15.1	7.4	<0.001
Other sugar drinks	21.6	9.9	<0.001	21.7	8.9	<0.001	21.7	10.0	<0.001
Candies and chocolates	2.7	2.1	0.138	2.4	1.7	0.017	2.5	1.9	0.008
Fried dough foods	7.2	5.3	0.002	7.5	5.4	0.013	7.4	5.5	<0.001
Pickles	7.5	11.7	<0.001	10.3	11.2	0.408	9.1	11.1	0.003
Processed meat	3.3	3.5	0.644	3.7	3.2	0.169	3.6	3.3	0.299
Pastries	14.7	13.4	0.246	13.8	13.0	0.329	14.2	12.7	0.029

Five controls were matched for each case according to the nearest neighbor principle, and the co-variates included in the calculation are age, BMI, education, smoking, alcohol use, water intake, tea consumption, and urinary pH.

**Table 2 nutrients-16-00214-t002:** Dietary patterns yielded by principal component analysis and their association with kidney stone disease.

Dietary Patterns	High Loadings (>0.3)	Low Loadings (<−0.3)	OR (95% CI) ^#^	*p*
Male				
PC1	Poultry, meat from other livestock, crustaceans *, pork *, tofu *		1.02 (0.98, 1.07)	0.327
PC2	Crustaceans *, freshwater fish *	Other sugary drinks, carbonated drinks, pure juices	0.83 (0.78, 0.89)	<0.001
PC3	Yogurt *, meat from other livestock *	Dark vegetables, fresh vegetables	1.14 (1.06, 1.24)	<0.001
PC4	Processed meat *, innards *, carbonated drinks *	Dairy, yogurt, fruits	0.95 (0.88, 1.04)	0.279
PC5	Nuts, pickles, pastries *	Fresh vegetables *, pure juices *	0.79 (0.72, 0.87)	<0.001
PC6	Marine fish, soymilk, tofu	Dark vegetables, candies and chocolates *, fresh vegetables *	0.98 (0.89, 1.08)	0.659
PC7	Fried dough foods *, soymilk *, dairy *	Eggs, carbonated drinks, other sugary drinks	1.05 (0.95, 1.15)	0.355
PC8	Rice, wheat, nuts	Pickles, potatoes	1.10 (0.99, 1.22)	0.085
PC9	Wheat	Rice	1.04 (0.94, 1.17)	0.432
Female				
PC1	Mushroom *, meat from other livestock *, tofu *, poultry *, crustaceans *		1.07 (1.03, 1.12)	0.001
PC2	Fresh vegetables, dark vegetables, freshwater fish *	Other sugary drinks, carbonated drinks *, pure juice *	0.88 (0.82, 0.93)	<0.001
PC3	Yogurt, fruits, dairy *	Pickles, innards	1.05 (0.98, 1.12)	0.141
PC4	Carbonated drinks, other sugary drinks, pork	Whole grains and mixed beans, potatoes *	1.11 (1.04, 1.20)	0.004
PC5	Crustaceans, nuts	Dark vegetables, dairy, fresh vegetables *	0.79 (0.72, 0.85)	<0.001
PC6	Carbonated drinks, marine fish, crustaceans *	Fried dough foods, nuts, candies and chocolates *	1.03 (0.95, 1.13)	0.445
PC7	Carbonated drinks, fried dough foods, other sugary drinks	Poultry, meat from other livestock, processed meat *	0.94 (0.87, 1.02)	0.149
PC8	Rice, poultry *, wheat *		1.12 (1.02, 1.22)	0.014
PC9	Nuts, pork *	Processed meat, soymilk, tofu *	1.00 (0.92, 1.09)	0.916
PC10	Rice, juice *, marine fish *	Wheat	0.93 (0.85, 1.01)	0.099

* Less than three food groups have absolute factor loadings greater than 0.3; food groups with factor loadings between 0.2 and 0.3 were selected. # Covariables adjusted in principle component regression: total energy, age, BMI, education, smoking, alcohol use, tea drinking, water intake, urinary pH.

**Table 3 nutrients-16-00214-t003:** Coefficients of least absolute shrinkage and selection operator regression and post-selection inference.

Food Items	OR (LASSO Regression ^#^)	Post-Selection Inference ^#^	
		OR	*p*
Male			
Pure juices	1.0047	1.0055	<0.001
Fried dough foods	1.0022	1.0057	0.127
Other sugary drinks	1.0019	1.0026	0.014
Meat from other livestock	1.0017	1.0043	0.040
Carbonated drinks	1.0006	1.0010	0.100
Soymilk	1.0001	1.0009	0.299
Fruits	1.0001	1.0005	0.216
Pickles	0.9891	0.9781	<0.001
Female			
Fried dough foods	1.0045	1.0063	0.045
Innards	1.0029	1.0053	0.121
Meat from other livestock	1.0024	1.0040	0.099
Pork	1.0019	1.0028	0.007
Mushrooms	1.0018	1.0030	0.090
Pure juices	1.0015	1.0019	0.086
Carbonated drinks	1.0014	1.0017	0.007
Soymilk	1.0013	1.0016	0.024
Other sugary drinks	1.0013	1.0015	0.033
Wheat	1.0011	1.0015	0.071
Poultry	1.0011	1.0015	0.499
Dairy	1.0007	1.0009	0.058
Yogurt	1.0007	1.0011	0.094
Rice	1.0002	1.0004	0.103
Crustaceans	0.9934	0.9904	0.001
Pickles	0.9961	0.9926	0.009
Eggs	0.9987	0.9968	0.117
Nuts	0.9989	0.9965	0.133
Freshwater fish	0.9996	0.9981	0.235

# Covariables adjusted in principle component regression: total energy, age, BMI, education, smoking, alcohol use, tea drinking, water intake, urinary pH.

## Data Availability

The data presented in this study are available on request from the corresponding author. The data are not publicly available due to privacy of the study participants.

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
