# Peer review of "Analysis of Dietary Patterns Associated with Kidney Stone Disease Based on Data-Driven Approaches: A Case-Control Study in Shanghai"

_nutrients, 2024, doi:10.3390/nu16020214_

Round 1
Reviewer 1 Report
Comments and Suggestions for Authors
The manuscript would be strengthened by hard data, such as:
1) stone composition of those who in your survey who had kidney stones (.e., oxalate, uric acid, cysteine, mixed composition, etc).
2) assessment/correlation of stone composition with diet reported
3) data needed on analysis of both serum and 24 hour urine findings in the stone population (i.e., Ca++, uric acid, urine pH, and specific urine findings of citrate, oxalate, urate, phosphate, total 24 hour urine volume -- and the correlation with what type of stone. High crustacean intake was protective in this study, and such an intake may be high in urate content -- a dichotomy that requires the urine uric acid results in both "high" and "low" shell fish consumers.
4) remove adjectives that have no meaning -- such as VERY, GENERALLY, MOST.
5) You state that a high beverage intake was a risk factor for stone formation and HYPOTHESIZE that high sugar and additives content was the causative factor (but no data presented to support that statement). No information about water consumption -- since a high water intake (urine output exceeding 2.5 liters/day) is noted to be a protective factor against stone disease. The higher the water intake correlates linearly with a decrease in the incidence of clinical stone disease.
Comments on the Quality of English LanguageUse of term "healthy and unhealthy" diets need to be defined by hard core biochemistry findings.
Reviewer 2 Report
Comments and Suggestions for Authors
Overall, this is a generally well done study of dietary patterns in kidney stone disease among a Chinese population using more detailed and advanced statistical analyses. Some items may improve the presentation of the data and the discussion of findings.
Title: It would be helpful to specify that the study was done exclusively in a Chinese population in the title.
Results: lines 162-170: Characterizing the PC2, PC3 and PC4 as “healthy” or “unhealthy” suggests bias. Alternative terms should be used at this point within the Results section. For example, line 172 “healthy low beverage pattern” could be described as “low beverage pattern.”
Supplementary Table 2: It would be helpful to the reader to bold type the values that are at or greater than 0.3.
Discussion: line 214: Again, “healthy low beverages pattern” would better be described using an attribute of the “type” of beverages in the analysis (e.g., low sugared beverages pattern”) as the analysis was adjusted for tea drinking which is a “healthy” (?) beverage. In other words, characterize the pattern more explicitly regarding the type of beverage as done with the meats and vegetables rather than “healthy” or “unhealthy” as this would be more helpful in guiding individuals with KDS.
Line 220: “Beverages were the most likely risk factor” is misleading as water or tea can be considered “beverages.” Please be more precise in describing the type of beverages associated with stones. This is clear in subsequent parts of the Discussion but should be termed precisely throughout so as to do justice to the analysis.
Lines 238-244: This is an interesting commentary. It would be of importance to note that increased urinary pH may benefit uric acid stone formers but not calcium oxalate or calcium phosphate stone formers. This may also be an issue regarding crustacean intake (lines 260ff) which increases uric acid. Thus, not all stone formers may be alike due to the type of kidney stone being formed. A comment or disclaimer regarding the fact that the subjects were not stratified based on types of stone should be included.
Comments on the Quality of English Language
The quality of English is good. The comments on description of categories of foods does not refer to language but rather to scientific characterization.
Reviewer 3 Report
Comments and Suggestions for Authors
The manuscript by Yifei Wang et al “Analysis of dietary patterns associated with kidney stone dis-2 ease based on data-driven approaches”, as state in the title, used data-driven approach to identify preventive- and risk factors for kidney stone disease in a sample of 6396 residents from a suburb of Shanghai. Their statistical approach accounts for interrelations between nutrients present in various kinds of foods. However, more care needs to be exercised in order to properly understand their findings.
As stated in the abstract, they found that “beverages” are a risk factor for developing KSD. The broad definition of a beverage includes fluids such as water, soda, etc. Therefore this finding is counterintuitive given that increasing water intake is often the advice given to patients who suffer from KSD. Fortunately, in the discussion and conclusion section the authors talk about “sweetened beverages”, not just “beverages”. It makes the findings more plausible. If the term “beverages” throughout the manuscript really refers to “sweetened beverages”, the latter term should be used throughout the entire manuscript. In such a case, the presented results will make more sense.
As water intake from all sources is crucial for the analysis, please provide more detailed description of what information was inferred from the FFQ. Was the amount of water consumed from foods also inferred? A reference to description of the FFQ is not enough (you cannot expect readers go and read the referenced manuscript). A brief description is a must.
The use of PCA is of merit, given that the intakes of various nutrients are highly correlated among each other. The components reveal very interesting food patterns, such as PC4 with high consumption of processed meat and sugary drinks. Processed meat is often considered as detrimental for health. Given your results, now I am wondering whether it is due to meet itself, or the combination with increased consumption of sugary beverages. Standard papers do not evaluate such associations, so their conclusions could be biased. Please discuss briefly in the discussion section.
Who performed physical examinations (line 96)? Please provide items obtained from urine tests (line 97). A full list could be included in the supplement.
Why tea consumption is used as a covariate? Was tea/water consumption not evaluate in the FFQ?
Does the item “yoghurt” in Table 1 refers to sweetened or non-sweetened yoghurt, or both? Same about dairy and soymilk? In Europe and in the North America these products are generally sweetened.
In what kind of dishes are pickles commonly used in the Shanghai region? Maybe consumption of other nutrients, not just pickles alone, is protective from KSD? Similarly, what do the authors understand as processed meat? Is it sweetened?
No information on PC5? (~line 177). Why pickles, nuts, and pastries intakes are combined in PC5? Are pickles with pastries served together in suburban Shanghai? Please explain with some examples. This associations is not straightforward to non-Asian readers.
Paragraph titles: ; 118 and 129 in italics, please..
Discussion:
PCA demonstrated that preferences for meat was related to high consumption of sugary drinks (PC5). This is an interpretation issue, but do you think that meat alone is a risk factor for KSD? The former studies seems to confirm this interpretation, but did they properly account for sugary drinks consumption? Maybe more studies are needed?
